# Analysis of the Elderly’s Preferences for Choosing Medical Service Facilities from the Perspective of Accessibility: A Case Study of Tertiary General Hospitals in Hefei, China

**DOI:** 10.3390/ijerph19159432

**Published:** 2022-08-01

**Authors:** Zao Li, Yanyan Gao, Li Yu, Charles L. Choguill, Weiyi Cui

**Affiliations:** 1College of Architecture and Art, Hefei University of Technology, Hefei 230009, China; lizao@hfut.edu.cn (Z.L.); yul@cardiff.ac.uk (L.Y.); cuiweiyi@mail.hfut.edu.cn (W.C.); 2School of Geography and Planning, Cardiff University, Cardiff CF24 3WA, UK; 3School of Public Affairs, Zhejiang University, Hangzhou 310018, China; cchoguill@gmail.com

**Keywords:** the elderly, distance, accessibility, preference, public health service, medical-seeking behavior

## Abstract

The accessibility of medical service facilities is a vital influence on elderly people choosing medical treatment. Encouraging residents to seek nearby medical treatment can facilitate the rational layout and diversion of urban medical facilities and reduce health resource waste. However, due to accessibility factors, elderly people may not choose the nearest hospital. In order to investigate such factors, we conducted a questionnaire survey among the elderly from 10 tertiary general hospitals in Hefei, China. On the basis of the origin-destination (OD) cost matrix analysis and statistical analysis of 830 valid questionnaires, this paper analyzed the elders’ selection rules when choosing medical facilities and the factors considered when making a choice. The study found that although 85% of elderly participants valued a short distance to tertiary hospitals, only 31% of them attended the closest hospitals in reality, which correlated with regularity according to their education level, travel activity status, and place of residence. The elderly highlighted road congestion, convenience of public transport stations, and number of transfers as critical in determining whether they sought nearby medical treatment. According to the results, effective ways to encourage the elderly to attend their nearest hospital, from the perspective of accessibility, include easing road congestion, improving the layout of public transport stations, and optimizing urban public transport routes. In particular, when planning future medical facilities, attention should be paid to the elderly with primary school education or below, who cannot travel independently, and those who live far from the city center.

## 1. Introduction

### 1.1. The Elderly and Their Medical Decisions

Medical service facilities have become an important part of China’s urban public health service over the last decade. The *Opinions on Formulating and Implementing Elderly Care Service Projects* report issued by the General Office of the State Council of the Chinese Government in 2017 indicates that it is necessary to support the elderly by developing elderly-friendly cities and improving elderly-oriented systems in the provision of urban public service facilities [1]. This will help to transform elderly-oriented public facilities and optimize the distribution of medical service facilities for the elderly in China.

With the improvement of the Chinese medical insurance system, the overall medical service level, and the number of elderly medical service facilities, there are now more options for elderly people seeking medical care [2,3,4,5]. Nevertheless, the medical decisions of elderly people are still dependent on their social and economic attributes. According to Kaambwa et al., elderly people living in remote areas tend to choose telehealth services due to the distance to a medical service facility [6]. It was suggested by Du et al. that elderly suburban residents are more likely to rely on cars to attend medical appointments than those living in the city center. In the city center, due to the greater accessibility of medical facilities, most elderly people choose to walk rather than drive to medical service facilities. This may raise a concern that the distribution of medical service facilities in the suburbs is relatively scattered, accounting for the greater use of cars for this purpose. It may also reflect the inadequacies in public transportation compared with that in the city center [7]. Research by Jana et al. suggests that when the rural elderly choose a public medical service facility, their family plays a crucial role as a family member often accompanies them. This companionship is important for two reasons; first, the elderly generally prefer someone to help with communication and discussion, and second, the family member provides companionship during the potentially long wait in public facilities [8]. The research of Teng et al. shows that gender, age, and education level are important determinants of emergency medical use by the elderly after accounting for predisposing factors, demands, and other relevant factors [9]. In conclusion, the social factors of the elderly, including their residence, household registration type (rural or urban residents), family structure, gender, age, and education background, are associated with their behaviors in seeking medical treatment.

It should be noted that the facility where the elderly seek medical advice and treatment is restricted by their spatial mobility due to physical conditions [10,11,12,13,14]. Their choice of hospital is often related to distance. Medically, distance correlates to the availability of medical resources and treatment convenience. Gao and Li found that there was a strong correlation between the elderly’s evaluation on “whether it is close to home” concerning medical facilities and their ultimate satisfaction evaluation; furthermore, the proximity of medical facilities was one of the reasons for the high-frequency visit by the elderly [15]. Love and Lindquist studied the accessibility of general and geriatric hospitals and found that the vast majority of metropolitan elderly residents within the state of Illinois, USA, live close to medical service facilities [16]. Yang et al. found that if elderly patients are far away from medical facilities, they substitute visits to pharmacies for outpatient visits to medical institutions where they may receive better care, which may have a negative impact on their health [11]. These studies show that for some elderly people, distance is of great significance in medical decision making.

In the academic literature, distance is an important indicator of accessibility. Some scholars define accessibility as a spatial barrier and operationalize it in terms of spatial distance and time cost [17,18,19,20]. A reasonable provision of medical facilities that increases accessibility has always been a major task in urban planning and development. The most common urban planning approach to enhancing medical accessibility is to reduce the distance between homes and medical facilities as much as possible [21,22]. Currently, Chinese residents are strongly encouraged to choose the closest medical facilities, thereby enabling a fair distribution of medical resources, rational diversion of patients, and resource and energy efficiency [23,24]. Considering the distance to medical services and understanding the preferences of residents for choosing them are critical in urban development.

Apart from the accessibility factor of distance, other factors are related to elderly patients’ decisions in choosing a medical institution: other accessibility issues (such as the number of transfers, travel time, and transportation options), the quality of available medical and nursing staff, and the level of the medical institution itself. This study investigates tertiary general hospitals, which are the highest level of medical facilities in China. They are medical institutions that provide medical and health services across counties (districts), cities, and provinces, including comprehensive medical care, teaching, scientific research, and public health services. The bed number, diagnosis and treatment subjects, medical equipment, and health technicians must correspond to the hospital’s service functions, technical level, and management requirements. China has a unified evaluation standard for such hospitals [25]. However, even in the face of equivalent medical facilities and the same quality of medical staff, the elderly may consider the convenience of transportation and transfers. Distance is just one of the factors considered, although it is an important one.

At present, most of the Chinese literature provides discussions on proximity of primary medical facilities [26,27]. In the *Implementation Plan for the Constructing Efficient and High-Quality Medical and Health Service System during the 14th Five-Year Plan*, it is mentioned that the elderly should be guaranteed high-level medical services within the proximity [28]. Therefore, it is necessary to study on the elderly’s behaviors in seeking medical treatment at a tertiary general hospital nearby. Such studies are currently rare in China.

### 1.2. Hypothesis and Aims of the Study

The addition of the accessibility variable complicates the medical facility selection of the elderly, who may be torn between attending the closest facility or the one that can provide the best treatment. Personal circumstances may also be considered, leading them to decide to attend a more distant facility rather than the nearest. Thus, it may not be possible to choose the nearest hospital. This leads to the present research hypothesis that most older people in China do not consider the nearest hospital when seeking medical care because proximity is less important than other accessibility factors. Our research aims to understand older adults’ considerations and strategies when making this trade-off, such as what factors are related to their behavior in choosing the closer tertiary general hospital. On this basis, we make practical suggestions for urban planning that promote choosing nearby medical treatment. The outcomes of this research provide a rational basis for locating urban medical service facilities and resources, particularly to meet the needs of the elderly, while also saving energy.

This paper addressed the following research questions: (1) How highly do the elderly evaluate the distance to tertiary general hospitals in relation to other considerations? (2) What type of elderly patients prefer the nearest facility for medical treatment? and (3) What factors do they take into account in reaching this decision? A questionnaire survey of elderly patients at Hefei tertiary general hospital was conducted. The data were analyzed by OD cost matrix model calculation, correspondence analysis, and regression analysis to answer the research questions.

## 2. Methodology

The case study is in Hefei, the capital city of Anhui Province, China. Hefei urban city is formed by four administrative districts, including districts of Luyang, Shushan, Baohe District, and Yaohai [29] (Figure 1). According to the existing city comprehensive plan, the scope of the urban center has been expanded from the First Ring Road to the Second Ring Road and continues to extend outwards along the ring roads [30].

At the end of 2017, there were 1198 medical facilities in Hefei (including tertiary hospitals, secondary hospitals, primary hospitals, community health service centers, community health service stations, and health clinics). According to the evaluation standard of tertiary hospitals in Anhui Province (see Appendix A Table A1 for details), there are 12 tertiary hospitals rated by the Hefei Municipal Government, of which 2 are specialized, and 10 are general hospitals [31]. The two specialized hospitals are not considered within this study’s scope in order to ensure the consistency of medical services included. The 10 general hospitals include 4 tertiary hospitals within the First Ring Road, 3 hospitals between the First and Second Ring Roads, and 3 outside the Second Ring Road. The distribution of hospitals is shown in Figure 2.

Admittedly, certain factors still lead to complexity in decision making, despite the accepted standard service provision of tertiary hospitals, including the geographical location of facilities, medical insurance, well-known experts, and the availability of specialized medical equipment. However, in contrast to the less well-equipped primary and secondary hospitals, tertiary general hospitals are seen as having more or less the same quality and level of service provision. Therefore, medical service standards are assumed to be uniform among all tertiary general hospitals. Other social attributes of tertiary hospitals are not considered in this analysis but are only discussed from the perspective of accessibility.

Our questionnaire survey was divided into two stages. The first consisted of one-to-one interviews conducted at hospitals, which occurred in the period of March–June 2018. The data were analyzed using statistical analysis in Statistical Product and Service Solutions (SPSS) and an OD cost matrix model in Geographic Information Systems (GIS) to investigate whether the elderly seek medical treatment at the nearest hospital and the rules they follow when making such decisions. This helps to explore the behavior regularity of choosing nearby medical treatment in relation to elderly people’s attributes. For scientific purposes, the subjects had been followed since 2018. In addition, as the second stage of the survey, on the premise of knowing whether or not they went to the nearest hospital, we conducted follow-up telephone interviews to them during January–June 2021 to learn which accessibility factors were considered.

### 2.1. One-to-One Interviews at the Hospital

The sample size for the one-to-one interviews conducted in 2018 was determined by referring to the number of visits by the elderly to tertiary general hospitals in Hefei urban city in the previous year. According to 2017 statistics from the Hefei Municipal Statistics Bureau and Hefei Municipal Health Commission, there were 2.75 million annual visits by the elderly to these hospitals [32]. The formula of sample size estimation is
(1)$$\mathrm{Sample}\;\mathrm{size}=\frac{\frac{z^{2}\times p\left(1-p\right)}{e^{2}}}{1+\left(\frac{z^{2}\times p\left(1-p\right)}{e^{2}N}\right)}$$

In the formula, *N* is the total visit number of the elderly (2.75 million), *e* refers to the margin of error (5%), *p* is the response distribution (50%), confidence level is 95%, and z-score is 1.96 [33,34]. A minimum sample size of 384 patients was calculated. During the survey, 100 subjects were investigated in each of 10 hospitals. The subjects were aged ≥60 years and were randomly selected outpatients. Hospital staff did not interfere with the survey, so the objectivity and authenticity of the respondents’ answers can be ensured. Consent was obtained from each elderly before the interviews. Questionnaires with missing answers, or too short filling time, were deemed invalid. According to the survey, it was found that some elderly people were from areas outside Hefei urban city and even Anhui Province. For these elderly people, the reasons for selection of hospitals were more complicated, while accessibility of the hospital was generally not one of them [35,36]. Since this study provides a discussion on the elderly’s choice of hospitals from the perspective of accessibility, such data are not helpful for the survey and should be excluded. Finally, 830 valid questionnaires were collected, satisfying the minimum sample size of 384. The numbers of valid and excluded questionnaires from each hospital are shown in Table A2 of the Appendix A. The questionnaire contained information about the basic attributes of gender, age, household registration type, number of people living at home, number of generations living at home, education level, original occupation, number of children, household per capita annual income, type of medical insurance, travel activity status, and number of private cars owned. In addition, the items also included each participant’s residence location, as well as their preference for hospital distance. The participants’ residence locations were entered into the GIS software in Figure 2. However, the information about whether or not the elderly chose the nearest hospital was not available in the first stage of questionnaire survey. For these elderly people, they lacked quantitative knowledge to accurately identify whether the hospital that they went to was the nearest. Hence, the OD cost matrix analysis was needed.

### 2.2. OD Cost Matrix Analysis

The distance from each participants’ residence to the selected hospital and the nearest hospital can be obtained using the OD cost matrix model analyzed by the GIS network. As a component of GIS network analysis, the OD cost matrix model is often used to find and measure the lowest cost path from multiple source points to multiple destinations in the network. In this research, the OD cost matrix model was able to quickly solve the large matrix *M* × *N* (*M* refers to the set of origins and *N* refers to the set of destinations) problem and record the minimum distance between each source point and each end point. There are two advantages of this model. One is that it allows each of the elderly to freely specify the number of destinations that they had considered in making their hospital decision. The other one is to consider the road network in reality. In ArcGIS, the OD cost matrix solver can only output straight lines but cannot display the actual shape of the path. However, this does not affect the analysis result because the distance cost stored in the “line” attribute table reflects the distance of the road network rather than the straight-line distance [37,38,39,40,41]. Since only participants in Hefei were selected in the survey, the model established the road network data set on the basis of the road network of Hefei and its surrounding areas, and then created the layer of OD cost matrix. By setting the urban boundary of Hefei as the boundary of the maximum search scope, straight lines between the homes of participants (the origin points) and all 10 hospitals (the endpoints) can be generated in the model, including the lines between the residence of each elderly person and the nearest hospital, as well as the visited hospital (Figure 3). The road network distance in the attribute sheet of each line was counted, and the distance between the residence and the chosen hospital was screened out to obtain the actual medical distance travelled by the elderly patients. The distance for each participant and their nearest hospital was screened through Matlab. Assume that the distance matrix of origin and destination obtained by OD cost analysis is ${D=(d}_{\mathrm{ij}}{)}_{M\;\times \;N}$, where *i* is the elderly, *j* is the hospital, *M* refers to the set of origins (i.e., the residences), and *N* refers to the set of the destinations (i.e., the hospitals). The distance from where each elderly *i* is to the nearest hospital is
(2)$${\bar{d}}_{i}{=\min\{d}_{i1}{,d}_{i2},\cdots {,d}_{\mathrm{in}}\},\;i=1,2,\cdots ,m.$$

In this formula, *m* is the total number of residences and *n* is the total number of hospitals. Finally, the program calculated which hospital was the nearest for each elderly person, as well as the distance between the residence of each elderly and the nearest hospital. By comparing the actual medical distance of the elderly with the closest medical distance, we can determine whether they obtained medical treatment nearby.

### 2.3. Telephone Follow-Up Interviews

The purpose of the telephone follow-up interview was to find out what factors were related to the elderly’s behavior of seeking medical treatment nearby. During the survey, both elderly participants who did and did not go to their nearest hospitals were included. Before the telephone follow-ups, the nearest hospitals to the participants were determined through the GIS model and Matlab. The participants were asked to evaluate the accessibility of the nearest hospitals they went to or would go to, including the mode of transportation needed or usually used, as well as details such as travel time, cost, and the number of transfers required for each mode of transportation. In this way, the accessibility factors of the nearest hospitals that encourage or hinder their access would be explored.

Each evaluation index contained specific evaluation criteria. For example, for elderly patients who needed to take public transport to their nearest hospital, further evaluation items under the time category included the time spent from home to the public transportation station, waiting time, riding time, transfer time, time spent from the station to the hospital, and other required timings. Other evaluation indicators and evaluation methods are shown in Table 1 and its note below.

### 2.4. Statistical Analysis

IBM SPSS Statistics 25 was used for input and analysis of the result at the two survey stages (one-to-one interview and telephone interview). The data were entered by two people and cross-checked.

(1)In order to analyze the preference for hospital distance among the elderly with different attributes, the chi-squared test was performed on their basic attributes of the elderly and perceived importance of distance. The elderly’s preference for distance, or, namely, the importance of distance, was used as the dependent variable. The elderly’s attributes of gender, age, household registration type, number of people at home, number of generations at home, education background, original occupation, number of children, household per capita annual income, medical insurance type, travel activity status, residence place, and number of private cars owned were used as independent variables (see Appendix A Table A3 for assignment of variables). Upon the chi-squared test, statistically significant attributes (*p* < 0.05) were screened out. Then, multiple correspondence analysis was performed again on these significant attributes and the preference for distance.

Correspondence analysis is similar to the variable dimension reduction analysis method of principal component analysis. It is primarily used for the qualitative analysis of two-dimensional or multidimensional contingency table data. It can analyze the correlations between variables and the difference or similarity between the same variable categories and characterize the corresponding relationships using graphs. Correspondence analysis was carried out on the basis of chi-squared statistics, with the result depicted as a scatterplot. The variable correlation was represented as the spatial position relationship of the scattered points.

(2)In order to analyze the regularity of the elderly’s behaviors in seeking medical treatment nearby, the chi-squared test was also firstly performed on the data. The elderly’s actual behaviors, or, namely, whether or not they chose the nearest hospital, was used as the dependent variable, and the attributes of the elderly were used as independent variables (see Table A3 for assignment of variables), thus identifying the relationship between the elderly’s attributes and their behaviors in seeking medical treatment.(3)In order to find out the factors associated with the elderly’s behaviors in seeking medical treatment nearby, statistical analysis was conducted for the elderly’s evaluation on accessibility under different means of transportation and their actual behaviors (whether they chose the nearest hospital) on the basis of the data obtained from telephone interviews. The assignment of variables are referred to in Table A3. After the statistically significant accessibility factors were screened out through the chi-squared test, regression analysis was performed again to finally identify the accessibility factors related to the elderly’s actual behaviors.

## 3. Results and Analysis

### 3.1. Elderly Preferences for Distance

Table 2 shows the statistics of elderly attributes from the questionnaire survey. In addition, the survey data revealed the distance preferences of elderly patients (Table 3). By referring to the variable assignment of distance importance, we made statistics on the data in Table 3 and found through calculations that the mean value was 2. That is, the elderly’s average preference for distance is “important”. The standard deviation was 0.6. It was found that more than 85% of the elderly believed that hospital distance was important to them to some extent. Most older people valued their proximity to tertiary hospitals and were eager to seek medical treatment nearby.

To further analyze the distance preferences of the various elderly patient sub-groups, their attributes and the results shown in Table 3 were analyzed using correspondence analysis in SPSS. Given the low proportions of “unimportant” or “very unimportant” responses, these two responses were combined as “neutral or not important”.

The attributes of the elderly and their preference for distance were subjected to a chi-squared test, and the results are shown in Table 4. According to the analysis, the statistically significant attributes were household registration type, education level, original occupation, household per capita annual income, and type of medical insurance. Then, these attributes were further subject to correspondence analysis against the elderly’s perceived importance of distance. The results showed that the inertia contribution rates in the first and second dimensions were 52.5% and 27.7%, respectively. The cumulative contribution rate was 80.2%, which was greater than 75%. Therefore, it is reasonable and adequate to use the first two dimensions for analyzing the relationship between row and column variables [42].

Figure 4 shows the diagram of correspondence analysis, where the relationship between the variables can be seen intuitively. The closer the distance between scattered points, the more obvious the associated tendency. It can be seen from the figure that the elderly individuals holding Hefei urban household registration (hukou); working in public institution or enterprises before retirement; enjoying medical insurance for being retired cadres, urban workers, or residents; with a household per capita annual income ≥ CNY 10,000; or having a junior middle school education or above tended to think that tertiary hospitals’ distance was “very important” and “important”. Elderly people with Hefei rural or non-Hefei household registration, self-employed or unemployed or working as farmers before retirement, having new rural cooperative medical insurance or having paid their own costs, with a household per capita annual income < CNY 10,000, or having a primary school or below education were more inclined to think that the distance was “neutral or not important”. These results suggest an important question: What type of older patient primarily chooses the nearest tertiary hospital?

### 3.2. Regularity of Choosing Nearby Medical Treatment for Older People

The relationships between a distance preference and the attributes of elderly participants were analyzed using chi-squared tests (Table 5). According to the survey data and analysis, more than 85% of elderly patients valued a short distance between their home and tertiary hospitals. However, the chi-squared results for the actual behavior of the elderly and their distance preference were not statistically significant. The distance preference was not correlated with their hospital selection. In other words, although the majority of the elderly valued a short hospital distance, their initial intention of going to the nearest hospital was often dominated by other objective factors. They had to compromise according to the actual conditions, leading to a gap between their preference and actual behavior. The data in Appendix A Figure A1 shows the outcomes of where participants chose to undergo medical treatment. When a participant’s travel distance was the same as the nearest hospital distance, it indicates that they chose nearby medical treatment. It was found that among the 830 elderly people, 569 did not attend the nearest hospital, while the other 261 did. Only 31% of the elderly considered the nearest medical service facilities, while the other 69% abandoned the nearest one, even if it was a tertiary hospital. Thus, the proximity of hospitals was not attractive. To explore the characteristics of the elderly who chose the nearest medical treatment, a chi-squared analysis of their attributes and behavior was conducted. It shows that education level, travel activity status, and residential location were correlated with actual behavior. The results in Table 5 demonstrate that as the participant’s education level increased, a higher proportion of participants travelled past their nearest hospital. Compared with older people who cannot travel independently, older adults who travel independently but may still need auxiliary devices had a higher proportion for giving up the nearest hospital. Those who can travel independently without assistance had the highest rate of bypassing their closest hospital. This infers that the more mobile older people were more likely to give up the nearest medical treatment. Finally, those older people who live closer to the city center were more likely not to use the nearest hospital for medical treatment.

### 3.3. Association between Hospital Accessibility Factors and Elderly Behavior

As previously discussed, distance had little to do with the choice of hospital for the elderly. In addition, more than two-thirds of the elderly bypassed the nearest hospital and chose a more distant one. Through the telephone follow-up survey data, we obtained a multifaceted evaluation of the accessibility factors of the nearest hospital. Then, the promotion and inhibition factors of choosing the nearest hospital were able to be summarized. Chi-squared analysis was conducted on the accessibility evaluations of elderly patients and whether they chose the nearest hospital or not. The results in Table 6 confirm that the factors of “transfer number”, “station convenience”, and “road congestion” for public transportation and mixed mode, as well as the “road congestion” factor for private vehicles, non-motor vehicles, and walking mode, were significantly related to the seeking of nearby medical treatment by the elderly. These above significant factors were taken as independent variables to conduct regression analysis with the behavior of the elderly; the results are shown in Table 7. For elderly patients that primarily used public transport, the number of transfers was correlated with their behavior of choosing nearby medical treatment (*p* = 0.000). The regression coefficient is positive, meaning that when a greater number of transfers is required to reach the nearest hospital, that hospital will be less likely to be chosen. In addition, the convenience of public transport stations was also correlated with their behavior (*p* = 0.016). The results show that the less convenient the station is, the more likely the nearest hospital is to be bypassed. Road congestion was also an associated factor (*p* = 0.034). The more congested the road to the nearest hospital is, the more likely it will be bypassed. Similarly, for elderly people using private vehicles, non-motor vehicles, mixed vehicles, or walking, traffic congestion was also associated with their behavior. The less congested the roads are, the more likely the nearest hospital will be chosen. The number of transfers and the convenience of the station were also correlated with whether elderly users of mixed-mode transportation would choose their nearest hospital. It was easier for them to bypass the nearest hospital if there were more frequent transfers and fewer convenient stations.

The farther away the elderly lived from the city center, the higher the proportion that sought treatment at their closest medical facilities. This phenomenon can be explained from the perspective of road congestion. As mentioned above, more straightforward navigation to the closest hospital promoted the seeking of medical treatment nearby. This is because urban road congestion becomes gradually alleviated further away from the city center. Additionally, in the investigation and interview processes, it was reported that most elderly participants with higher education levels either worked in public institutions before retirement or had professional employment. Many were still residing in unit housing in the old city or nearby, experiencing a congested road environment. In contrast, the majority of respondents with a low level of education were farmers. The majority of them lived in the city’s fringe areas outside the Second Ring Road with good road accessibility and, consequentially, they were more likely to seek nearby medical treatment.

## 4. Conclusions

By considering and contrasting the spatial distribution of Hefei tertiary general hospitals and the residential locations of the respondents, together with the questionnaire survey data, this research found that although the majority of elderly patient regarded the distance to tertiary hospitals as important, only about one-third of them sought medical treatment nearby. Hence, there was no correlation between the elderly’s subjective concern on the distance and their actual behavior. However, further exploration indicates that the accessibility of the nearest hospitals, according to traffic congestion, location of public transportation station, and number of transfers, is closely related to the decision of whether or not to seek medical treatment nearby, rather than distance. From an urban planning perspective, we argue that the most effective way to encourage nearby medical treatment is to alleviate road congestion, adjust the location of public transportation stations to increase convenience, and optimize public transportation lines to reduce the number of transfers required, rather than simply shortening the distances between hospitals and residential areas.

Urban traffic is a critical component in the development of a city’s functional layout. Traffic congestion has become an essential restrictor of urban development. It directly affects the quality of residents’ lives and travel and generates many social problems. This research confirms that severe road congestion in the urban center, such as within the First Ring Road, seriously impacts the daily lives of vulnerable groups, including the elderly who have difficulty traveling, and discourages them from visiting the nearest hospital, even if they can walk there. Improving the urban medical treatment environment and medical services should be considered from many aspects, including alleviating traffic congestion. Field investigations revealed that some of the causes of road congestion in downtown Hefei include disorderly parking and inadequate traffic organization. Initiation of a traffic improvement project in the Hefei city center would be an effective and immediate strategy for helping the elderly utilize the closest medical treatment facilities.

Public transport is an important part of the overall urban transportation system. Its development is an important symbol of a city’s degree of modernization and is the best way to solve urban traffic congestion. During the interviews, many participants mentioned that the public transportation lines in Hefei were insufficient, requiring them to transfer several times to reach a hospital, even close ones. The results of the study show that the Hefei public transport network does not adequately meet the needs of its residents. A well-functioning urban public transport system requires theoretical and technical support. Establishing an applicable urban public transportation network model and then determining the optimal travel paths will have a positive impact on improving the overall efficiency of the public transportation system and the convenience of daily travel, such as for medical treatment.

In line with the outcomes of this research, we suggest that it is critical to optimize the locations of public transport stations, although this is a multi-perspective problem. We argue that a reasonable station layout can reduce the investment cost of road construction and increase passenger convenience, especially for vulnerable groups such as the elderly. Most existing stations in China, including in Hefei, are planned by the urban road planning department according to station setting norms. However, this research suggests that station locations and passengers’ choice behavior should be interactive. The planning of stations needs to consider the relationship between public transport lines and the surrounding residents.

The outcome of this research raises a concern about social equality and providing public services for all. In line with the survey outcomes, elderly people with greater education or living closer to the city center were more likely to bypass the nearest hospital. This is because that these categories of the ageing population may be of governmental official, high-skilled laborers or those who used to work for state own enterprises and/or research institutions. They have more options in selecting hospitals. However, the elderly with a lower education level, e.g., elementary school or below, those living far from the city center due to the cost of housing, or those with low mobility that restricts independent travel, have to select the nearest hospitals. As vulnerable groups, they may not have more options. On the basis of the outcomes of this research, we would suggest that in the planning stage for the provision of medical facilities, the distributions of these groups should be considered. The distance between them and any newly built medical facility should be lessened to meet their high demand for seeking nearby medical treatment.

Overall, the findings of this study have clarified the rules of the elderly’s medical behavior and the factors considered when choosing a hospital. Moreover, instead of shortening the distance between hospitals and residential areas to encourage the elderly to seek medical treatment nearby, it would be more effective to address issues such as traffic congestion, inconvenient platform settings, and the high number of transfers required for public transportation. This observation has important implications for urban planning and medical care for the elderly in China. Although this study took place in Hefei City, the findings have reference significance for other similarly sized cities in China. Specifically, the description of elderly people with high demand for proximity to hospitals can be applied to other cities. Their distribution should also be considered in the future urban planning of these cities. Under the background of nationwide promotion of nearby medical treatment, the identified measures that promote nearby medical treatment can be promoted in other cities. China has experienced rapid urbanization over the past few decades. Relevant standards and norms may have been lagging behind the social and economic changes in Chinese society. For example, existing urban planning norms mainly consider land use and population ratio in the provision of public medical services but ignore the demands of gender, age, and socio-economic conditions. Moreover, the construction of public services cannot keep pace with urban expansion, so the urban transport system is in urgent need of renewal. Those are common problems in most Chinese cities. More critically, the ageing population has been the main issue of Chinese society. However, the demands of the elderly, especially accessibility, have not yet been considered key elements in planning for the provision of health and medical service facilities as a component of urban development. The proposals and suggestions in this research will contribute to academic debate and professional practice in Chinese urban planning and development, having strong social significance and practical value.

In our research, the one-to-one interviews to the elderly were conducted before COVID-19, and the telephone follow-ups were about their behavior and evaluation before the epidemic. COVID-19 was not considered in the scope of this study; however, it does have a certain degree of influence on elderly patients’ medical decisions, personal consciousness, and concept of health. Future studies could conduct surveys on elderly hospital patients to compare their medical decisions before and after the pandemic and determine the adjustments that they need to make with regard to choosing medical facilities in the context of epidemic prevention and control. In addition, future studies should no longer be limited to tertiary hospitals, but should also include other levels of medical facilities. Since the services provided by low-level medical facilities can be replaced by those of high-level medical facilities, including more levels of medical facilities, future research will more accurately capture the decision-making process of elderly patients choosing a hospital to visit.

## Figures and Tables

**Figure 1 ijerph-19-09432-f001:**
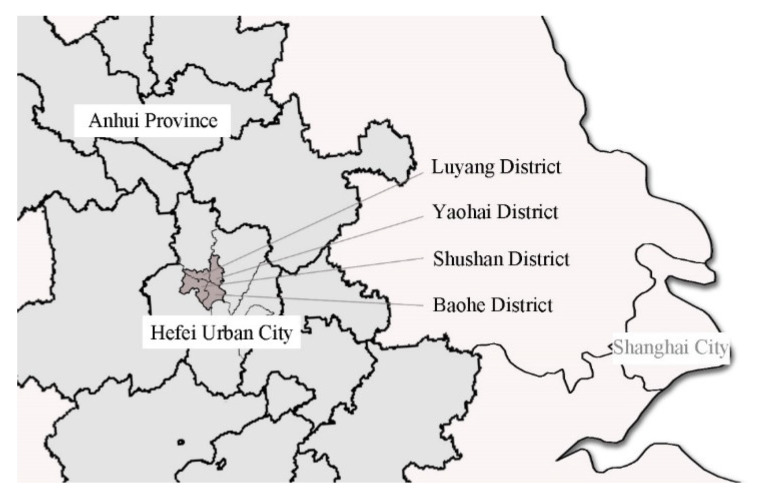
Location of Hefei urban city.

**Figure 2 ijerph-19-09432-f002:**
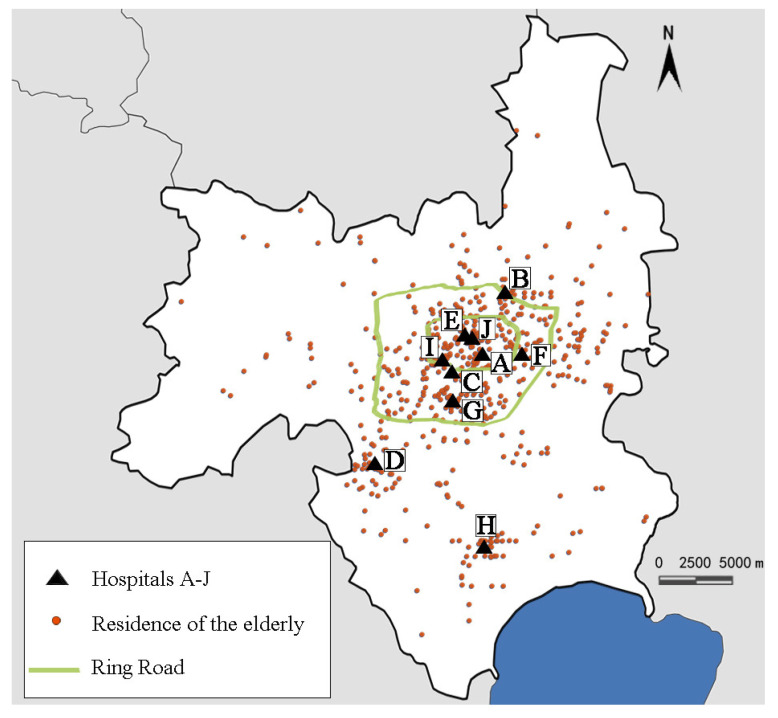
Location of hospitals and residences of the elderly in the survey.

**Figure 3 ijerph-19-09432-f003:**
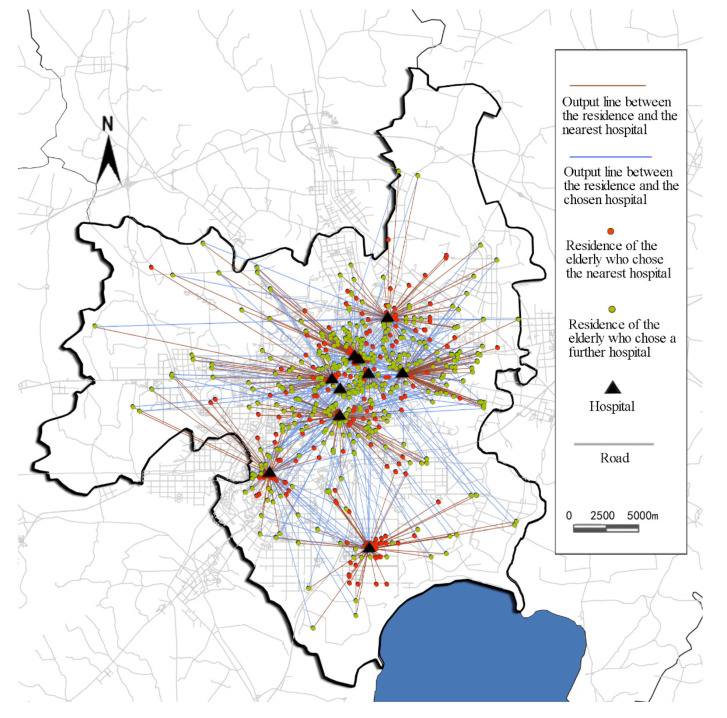
Output lines from the elderly’s residences to the nearest and the visited hospitals according to the OD cost matrix model.

**Figure 4 ijerph-19-09432-f004:**
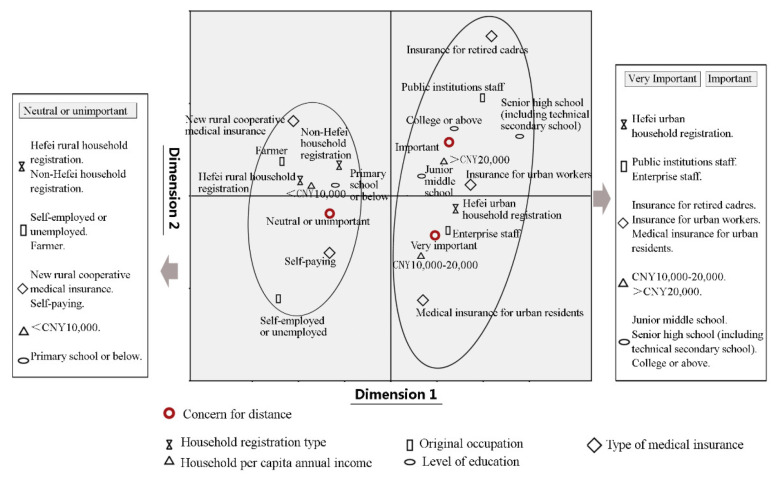
Correspondence analysis of the attributes of the elderly and their preference for distance.

**Table 1 ijerph-19-09432-t001:** Setting and criteria of evaluation index.

Means of Transportation	Category of Accessibility Evaluation	Evaluation Index	Criteria of Evaluation Index
Public transportation	Total time	Time (min)	Time spent from home to the station/waiting time/riding time/time for transfer/time spent from the station to the hospital/other required timings
	Total cost	Cost (CNY)	Fare/other required expenses
	Number of transfers	N	Number of transfers
	Convenience of station	Convenient/neutral/inconvenient	Distance from home to the station/distance from the station to the hospital
	Road congestion	Not congested/neutral/congested	Number of intersections and waiting time at intersections/number of passable lanes on the way/density of vehicles on the road/traffic order on the way
	Barrier-free facilities on the way	Complete/neutral/incomplete	Barrier-free setting on the way from home to the station/setting in the station/setting on the vehicle/setting on the way of transfer/setting on the way from the station to the hospital
Private motor vehicle	Total time	Time (min)	Time spent from home to the parking lot/driving time/time spent in the process of parking/time spent from the parking lot to the hospital/other required timings
	Total cost	Cost (CNY)	Fuel cost/toll/parking cost/other required expenses
	Road congestion	Not congested/neutral/congested	Number of intersections and waiting time at intersections/number of passable lanes on the way/density of vehicles on the road/traffic order on the way
	Convenience of parking	Convenient/neutral/inconvenient	Number of parking spaces/distance between the parking lot and the hospital
	Barrier-free facilities on the way	Complete/neutral/incomplete	Barrier-free setting in the parking lot/setting on the way from the parking lot to the hospital
Non-motorized vehicle	Total time	Time (min)	Time spent from home to the parking lot/riding time/time spent in the process of parking/time spent from the parking lot to the hospital/other required timings
	Total cost	Cost (CNY)	Rental fee for the vehicle/electricity or other consumption expenses/parking cost/other required expenses
	Road congestion	Not congested/neutral/congested	Number of intersections and waiting time at intersections/number of passable lanes on the way/density of vehicles on the road/traffic order on the way
	Convenience of parking	Convenient/neutral/inconvenient	Number of parking spaces/distance between the parking lot and the hospital
Mixed-mode of transportation	Total time	Time (min)	Time spent from home to the station or parking lot/waiting time/riding or driving time/time for transfer/time spent in the process of parking/time spent from the station or parking lot to the hospital/other required timings
	Total cost	Cost (CNY)	Fare/fuel, electricity or other consumption expenses/toll/rental fee for the vehicle/parking cost/other required expenses
	Number of transfers	N	Number of transfers
	Convenience of station	Convenient/neutral/inconvenient	Distance from home to the station/distance from the station to the hospital
	Road congestion	Not congested/neutral/congested	Number of intersections and waiting time at intersections/number of passable lanes on the way/density of vehicles on the road/traffic order on the way
	Convenience of parking	Convenient/neutral/inconvenient	Number of parking spaces/distance between the parking lot and the hospital
	Barrier-free facilities on the way	Complete/Neutral/Incomplete	Barrier-free setting on the way from home to the station/setting in the station/setting on the vehicle/setting on the way of transfer/setting on the way from the station to the hospital/setting in the parking lot/setting on the way from the parking lot to the hospital
Walking	Total time	Time (min)	Time spent on the way
	Total cost	Cost (CNY)	Expenses on the way
	Road congestion	Not congested/neutral/congested	Number of intersections and waiting time at intersections/density of pedestrians on the road/traffic order on the way
	Barrier-free facilities on the way	Complete/neutral/incomplete	Barrier-free setting on the way

Note: Mixed-mode of transportation includes public transportation + non-motorized vehicles, or public transportation + motor vehicles, or motor vehicles + non-motorized vehicles, or public transportation + motor vehicles + non-motorized vehicles. The value of time, cost, or transfer times is the accumulated value according to the evaluation index criteria in the rightmost column. The evaluation on convenience of station, road congestion, convenience of parking, and barrier-free facilities were filled in by the elderly on the basis of their actual feelings after the researcher explained the evaluation index criteria in the rightmost column.

**Table 2 ijerph-19-09432-t002:** The number and proportion of the elderly with various attributes.

Variable	Values	Number (N)	Percentage (%)	Variable	Values	Number (N)	Percentage (%)
Gender	Male	486	58.6	Number of children	None	9	1.1
	Female	344	41.4		1	137	16.5
Age	60–70	352	42.4		2	335	40.4
					≥3	349	42.0
	71–80	320	38.6	Household per capita annual income	<CNY 10,000 (<USD 1480)	246	29.6
	>80	158	19.0		CNY 10,000–20,000 (USD 1480–2960)	252	30.4
Household registration type	Hefei urban household registration	445	53.6		>CNY 20,000 (>USD 2960)	332	40.0
	Hefei rural household registration	218	26.3	Type of medical insurance	Medical insurance for retired cadres	60	7.2
	Non-Hefei household registration	167	20.1		Medical insurance for urban workers	278	33.5
Number of people living at home	1	89	10.7		Medical insurance for urban residents	171	20.6
	2	428	51.6		New rural cooperative medical insurance	297	35.8
	≥3	313	37.7		Self-paying	24	2.9
Number of generations living at home	One generation	512	61.7	Travel activity status	Travel independently without the aid of auxiliary devices	335	40.4
	Two generations	87	10.5		Travel independently with the aid of auxiliary devices	326	39.3
	Three or more generations	231	27.8		Cannot travel independently	169	20.4
Level of education	Primary school or below	427	51.4	Place of residence	Within the First Ring	256	30.8
	Junior middle school	281	33.9		Between the First and the Second Ring	343	41.3
	Senior high school (including technical secondary school)	79	9.5				
	College or above	43	5.2		Outside the Second Ring, within the urban area	231	27.8
Original occupation	Public institutions staff	168	20.2				
	Enterprise staff	256	30.8	Number of private cars owned	None	329	39.6
	Self-employed or unemployed	135	16.3		1	420	50.6
	Farmer	271	32.7		≥2	81	9.8

**Table 3 ijerph-19-09432-t003:** Statistics of the elderly’s preference for distance.

Distance Importance (Variable Assignment)	Number (N)	Percentage (%)
Very important (1)	115	13.9
Important (2)	591	71.2
Neutral (3)	113	13.6
Unimportant (4)	10	1.2
Very unimportant (5)	1	0.1
Total	830	100.0

**Table 4 ijerph-19-09432-t004:** Chi-squared test of elderly attributes and their preference for distance.

	Variable	Values	Importance of Distance N (%)			χ^2^ Value (*p*-Value)
			Very Important	Important	Neutral or Not Important	
Elderly attribute	Gender	Male	74 (15.2)	342 (70.4)	70 (14.4)	−0.445 (0.656)
		Female	41 (11.9)	249 (72.4)	54 (15.7)	
	Age	60~70	47 (13.4)	254 (72.2)	51 (14.4)	9.995 (0.119)
		71~80	54 (16.9)	226 (70.6)	40 (12.5)	
		>80	14 (8.9)	111 (70.2)	33 (20.9)	
	Household registration type	Hefei urban household registration	71 (16.0)	334 (75.0)	40 (9.0)	**17.240 (0.000)**
		Hefei rural household registration	30 (13.8)	140 (64.2)	48 (22.0)	
		Non-Hefei household registration	14 (8.4)	117 (70.1)	36 (21.5)	
	Number of people living at home	1	17 (19.1)	58 (65.2)	14 (15.7)	0.360 (0.835)
		2	61 (14.3)	315 (73.6)	52 (12.1)	
		≥3	37 (11.8)	218 (69.6)	58 (18.6)	
	Number of generations living at home	One generation	78 (15.2)	369 (72.1)	65 (12.7)	2.726 (0.256)
		Two generations	6 (6.9)	71 (81.6)	10 (11.5)	
		Three or more generations	31 (13.4)	151 (65.4)	49 (21.2)	
	Level of education	Primary school or below	49 (11.5)	296 (69.3)	82 (19.2)	**12.013 (0.007)**
		Junior middle school	43 (15.3)	209 (74.4)	29 (10.3)	
		Senior high school (including technical secondary school)	18 (22.8)	49 (62.0)	12 (15.2)	
		College or above	5 (11.6)	37 (86.1)	1 (2.3)	
	Original occupation	Public institutions staff	37 (22.1)	119 (70.8)	12 (7.1)	**31.919 (0.000)**
		Enterprise staff	30 (11.7)	200 (78.1)	26 (10.2)	
		Self-employed or unemployed	27 (20.0)	93 (68.9)	15 (11.1)	
		Farmer	21 (7.7)	179 (66.1)	71 (26.2)	
	Number of children	None	1 (11.1)	6 (66.7)	2 (22.2)	7.139 (0.068)
		1	24 (17.5)	100 (73.0)	13 (9.5)	
		2	36 (10.7)	258 (77.1)	41 (12.2)	
		≥3	54 (15.5)	227 (65.0)	68 (19.5)	
	Household per capita annual income	<CNY 10,000	17 (6.9)	181 (73.6)	48 (19.5)	**41.138 (0.000)**
		CNY 10,000–20,000	51 (20.2)	171 (67.9)	30 (11.9)	
		>CNY 20,000	47 (14.1)	239 (72.0)	46 (13.9)	
	Type of medical insurance	Medical insurance for retired cadres	13 (21.7)	46 (76.7)	1 (1.6)	**19.107 (0.000)**
		Medical insurance for urban workers	26 (9.4)	218 (78.4)	34 (12.2)	
		Medical insurance for urban residents	31 (18.1)	115 (67.3)	25 (14.6)	
		New rural cooperative medical insurance	39 (13.1)	205 (69.0)	53 (17.9)	
		Self-paying	6 (25.0)	7 (29.2)	11 (45.8)	
	Travel activity status	Travel independently without the aid of auxiliary devices	47 (14.0)	234 (69.9)	54 (16.1)	2.868 (0.238)
		Travel independently with the aid of auxiliary devices	26 (8.0)	271 (83.1)	29 (8.9)	
		Cannot travel independently	42 (24.9)	86 (50.9)	41 (24.2)	
	Place of residence	Within the First Ring	33 (12.8)	197 (77.0)	26 (10.2)	2.904 (0.574)
		Between the First and the Second Ring	42 (12.2)	255 (74.3)	46 (13.5)	
		Outside the Second Ring, within the urban area	40 (17.3)	139 (60.2)	52 (22.5)	
	Number of private cars owned	None	49 (14.9)	209 (63.5)	71 (21.6)	3.014 (0.222)
		1	38 (9.0)	333 (79.3)	49 (11.7)	
		≥2	28 (34.6)	49 (60.5)	4 (4.9)	

**Table 5 ijerph-19-09432-t005:** Chi-squared analysis of the elderly’s preference, attributes, and actual behavior.

	Factor		Actural Behavior N (%)		χ^2^ Value (*p*-Value)
			Choosing the Nearest Hospital	Not Choosing the Nearest Hospital	
Distance preference	Importance of distance	Very important	35 (30.4)	80 (69.6)	3.102 (0.212)
		Important	195 (33.0)	396 (67.0)	
		Neutral or not important	31 (25.0)	93 (75.0)	
Elderly attribute	Gender	Male	157 (32.3)	329 (67.7)	0.401 (0.527)
		Female	104 (30.2)	240 (69.8)	
	Age	60~70	114 (32.4)	238 (67.6)	0.816 (0.665)
		71~80	102 (31.9)	218 (68.1)	
		>80	45 (28.5)	113 (71.5)	
	Household registration type	Hefei urban household registration	135 (30.3)	310 (69.7)	0.817 (0.665)
		Hefei rural household registration	69 (31.7)	149 (68.3)	
		Non-Hefei household registration	57 (34.1)	110 (65.9)	
	Number of people living at home	1	32 (36.0)	57 (64.0)	1.132 (0.568)
		2	135 (31.5)	293 (68.5)	
		≥3	94 (30.0)	219 (70.0)	
	Number of generations living at home	One generation	166 (32.4)	346 (67.6)	2.019 (0.568)
		Two generations	29 (33.3)	58 (60.7)	
		Three or more generations	66 (28.6)	165 (71.4)	
	Level of education	Primary school or below	155 (36.3)	272 (63.7)	**14.856** **(0.002)**
		Junior middle school	83 (29.5)	198 (70.5)	
		Senior high school (including technical secondary school)	17 (21.5)	62 (78.5)	
		College or above	6 (14.0)	37 (86.0)	
	Original occupation	Public institutions staff	49 (29.2)	119 (70.8)	3.515 (0.319)
		Enterprise staff	92 (35.9)	164 (64.1)	
		Self-employed or unemployed	39 (28.9)	96 (71.1)	
		Farmer	81 (29.9)	190 (70.1)	
	Number of children	None	2 (22.2)	7 (77.8)	2.688 (0.611)
		1	42 (30.7)	95 (69.3)	
		2	104 (31.0)	231 (69.0)	
		≥3	113 (32.2)	236 (67.8)	
	Household per capita annual income	<CNY 10,000	68 (27.6)	178 (72.4)	6.048 (0.196)
		CNY 10,000–20,000	91 (36.1)	161 (63.9)	
		>CNY 20,000	102 (30.7)	230 (69.3)	
	Type of medical insurance	Medical insurance for retired cadres	15 (25.0)	45 (75.0)	2.554 (0.635)
		Medical insurance for urban workers	85 (30.6)	193 (69.4)	
		Medical insurance for urban residents	52 (30.4)	119 (69.6)	
		New rural cooperative medical insurance	102 (34.3)	195 (65.7)	
		Self-paying	7 (29.2)	17 (70.8)	
	Travel activity status	Travel independently without the aid of auxiliary devices	89 (26.6)	246 (73.4)	**6.964** **(0.031)**
		Travel independently with the aid of auxiliary devices	109 (33.4)	217 (66.6)	
		Cannot travel independently	63 (37.3)	106 (62.7)	
	Place of residence	Within the First Ring	66 (25.8)	190 (74.2)	**10.781** **(0.005)**
		Between the First and the Second Ring	104 (30.3)	239 (69.7)	
		Outside the Second Ring, within the urban area	91 (39.4)	140 (60.6)	
	Number of private cars owned	None	95 (28.9)	234 (71.1)	2.854 (0.240)
		1	135 (32.1)	285 (67.9)	
		≥2	31 (38.3)	50 (61.7)	

**Table 6 ijerph-19-09432-t006:** Chi-squared analysis on the accessibility evaluation of the elderly and their behavior.

			Total Time	Total Cost	Number of Transfers	Convenience of Station	Road Congestion	Barrier-Free Facilities on the Way	
Public transportation	Whether or not to choose the nearest hospital	χ^2^ value	22.455	5.241	**39.816**	**37.082**	**16.009**	1.505	
		*p*-value	0.664	0.073	**0.000**	**0.000**	**0.042**	0.471	
Private motor vehicle			Total time	Total cost	**Road congestion**	Convenience of parking	Barrier-free facilities on the way		
	Whether or not to choose the nearest hospital	χ^2^ value	20.180	2.330	**12.294**	0.819	2.330		
		*p*-value	0.447	0.802	**0.002**	0.664	0.802		
Non-motorized vehicle			Total time	Total cost	**Road congestion**	Convenience of parking			
	Whether or not to choose the nearest hospital	χ^2^ value	11.538	5.764	**6.951**	3.030			
		*p*-value	0.241	0.674	**0.031**	0.220			
Mixed-mode of transportation			Total time	Total cost	**Number of transfers**	**Convenience of station**	**Road congestion**	Convenience of parking	Barrier-free facilities on the way
	Whether or not to choose the nearest hospital	χ^2^ value	9.787	17.969	**12.621**	**20.443**	**8.149**	3.587	5.116
		*p*-value	0.550	0.082	**0.002**	**0.009**	**0.017**	0.166	0.077
Walking			Total time	Total cost	**Road congestion**	Barrier-free facilities on the way			
	Whether or not to choose the nearest hospital	χ^2^ value	13.266	11.058	**8.873**	3.765			
		*p*-value	0.151	0.136	**0.012**	0.152			

**Table 7 ijerph-19-09432-t007:** Regression analysis on the accessibility evaluation and the elderly’s behavior.

Mode of Transportation	Variable	Regression Coefficient	OR Value	*p*-Value
Public transportation	Number of transfers	0.887	2.428	0.000
	Convenience of station	0.001	1.001	0.016
	Road congestion	0.401	1.494	0.034
Private motor vehicle	Road congestion	0.686	1.986	0.001
Non-motorized vehicle	Road congestion	1.022	2.777	0.025
Mixed-mode of transportation	Number of transfers	1.411	4.102	0.006
	Convenience of station	0.004	1.004	0.003
	Road congestion	1.173	3.231	0.012
Walking	Road congestion	0.795	2.215	0.006

## Data Availability

Some or all data or models that support the findings of this study are available from the corresponding author upon reasonable request.

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
