# Peer review of "Analysis of the Elderly’s Preferences for Choosing Medical Service Facilities from the Perspective of Accessibility: A Case Study of Tertiary General Hospitals in Hefei, China"

_ijerph, 2022, doi:10.3390/ijerph19159432_

Round 1

Reviewer 1 Report

Thank you for the opportunity of reviwing this work. In my opinion there are major issues with the manuscript in its current state which I will try to list below: 

- The manuscript focus on urbanistic dimensions lacking public health and epidemiological data and discussions, therefore I am not sure it is the ideal journal for it.  

- The hypothesis and aims of the study would better fit in the end of the Introduction. Consider also briefly defining the methodology in the last paragraph of the Introduction. 

- The assumption starting in line 140 "This is beacause..." lacks support and referencing

- In my opinion the Literature Review section contains information that could be better summarized in the introduction and discussion sections 

- the methodology is not clear to me and, in parts, I think it is because the Methods section lacks objectivity.

- Could you plase clarify  why did you do a second phase of surveys by phone?

- Exclusion and Inclusion criteria are not clearly defined. Also, could you please explain why did you exclude the patients from other regions? 

- Figure 4 would benefit of o more objective description 

Author Response

Comments of Reviewer 1 and responses from the authors

Dear professor,

Thank you very much for your help in processing the review of our manuscript entitled "Analysis of the Elderly’s Preferences for Choosing Medical Service Facilities from the Perspective of Accessibility: A Case Study of Tertiary General Hospitals in Hefei, China" (ijerph-1768250).

After the revision, all concerns/suggestions you raised have been addressed as follows:

  1. The manuscript focus on urbanistic dimensions lacking public health and epidemiological data and discussions, therefore I am not sure it is the ideal journal for it.

The responses:

Our study have discussed the elderly's medical seeking behaviour and its influencing factors. In this journal, there are some similar articles about the influencing factors of medical behaviour. For instance,

Yang et al. (2021) studied the medical treatment behaviour of the elderly in Shanghai, China, and explored the group characteristics and influencing factors. Jing et al. (2020) discussed the behaviour rules and influencing factors of hospitalized elderly people in Shandong Province. Chu et al. (2019) studied marital migrants and migrant workers in Taiwan and analysed the related factors of their medical treatment behaviours. Yan et al. (2019) discussed the impacts of the hierarchical medical system on national health insurance on the residents’ health seeking behaviour in Taiwan.

Similar to our research, these studies explore medical treatment behaviour through quantitative questionnaire survey, but they explore other behavioural influencing factors rather than the accessibility of medical facilities. Therefore, our research is in line with the direction of this journal and innovative to some extent.

Not only that, this journal also contains studies on urban planning.

Cheng et al. (2021) took Shanghai City as an example to measure the spatial accessibility of urban medical facilities, and put forward planning suggestions for different levels of medical facilities in the city. Rao et al. (2022) evaluated the fairness of urban green space accessibility and put forward development suggestions from the perspective of accessibility. Wan et al. (2022) explored the impact of accessibility of health services in cities on residents’ self-rated physical and mental health.

The above literatures are also similar to our research to some extent.

  1. The hypothesis and aims of the study would better fit in the end of the Introduction. Consider also briefly defining the methodology in the last paragraph of the Introduction.

The responses:

Thank you for your comments. After the revision, we have put the research the hypothesis and aims at the end of the Introduction. See Section 1.2 for details (Lines 115-135).

And in the last paragraph of the Introduction, the methodology is simply defined. See Lines 132-135 for details.

  1. The assumption starting in line 140 "This is because... "lacks support and referencing.

The responses:

Thank you for your reminding. After the revision, Line 140 "This is because..." and the corresponding reference have been deleted and replaced with another previous study. This new reference is also a research result published by our team. However, this study can better illustrate the point we want to express, that is, "Their choice of hospital is often related to distance "(Line 70).

The revised words are  " Gao & Li (2022) found that there was a strong correlation between the elderly’s evaluation on "whether it is close to home" concerning medical facilities and their ultimate satisfaction evaluation; furthermore, the proximity of medical facilities was one of the reasons for the high-frequency visit by the elderly". See Lines 71-75 for details.

  1. In my opinion the Literature Review section contains information that could be better summarized in the introduction and discussion sections.

The responses:

We have summarized the literature review contents in the Introduction after the revision. See Lines 43-85 for details.

  1. The methodology is not clear to me and, in parts, I think it is because the Methods section lacks objectivity.

The responses:

The methods involved in this study mainly include three categories: questionnaire survey (one-to-one interview at hospital, and telephone follow-up interview), OD cost matrix model analysis, and data statistical analysis.

  • For questionnaire survey

Firstly, it is common to conduct a questionnaire survey on medical treatment behavior. For example, Yang et al. (2021) studied the medical treatment behavior of 439 elderly people by means of questionnaire survey. Chu et al. (2019) studied the medical treatment behaviors of 753 marital migrants and migrant workers through questionnaire survey. All of them explored the factors associated with medical treatment behavior through questionnaire data.

Secondly, the minimum sample size of questionnaire was objectively calculated based on the annual medical visits of the elderly and a reasonable formula. The relevant words have been added in the revised manuscript:

"The sample size for the one-to-one interviews conducted in 2018 was determined by referring to the number of visits by the elderly to tertiary general hospitals in Hefei urban city in the previous year. According to 2017 statistics from the Hefei Municipal Statistics Bureau and Hefei Municipal Health Commission, there were 2.75 million annual visits by the elderly to these hospitals (Hefei Municipal Bureau of Statistics, 2018). The formula of sample size estimation is

                             (1)

In the formula, N is the total visit number of the elderly (2.75 million), e refers to the margin of error (5%), p is the response distribution (50%), confidence level is 95%, and z-score is 1.96 (Li, Liu &Liu, 2020; Wu, 2010). A minimum sample size of 384 patients was calculated."    See the Lines 219-229 for details.

Thirdly, we finally collected 830 valid questionnaires, much larger than the minimum sample size of 384. Considering that the sample size of this survey is large enough, the subjectivity of individual elderly people’s answers has little influence on the overall survey results.

Fourthly, during the collection of questionnaires, "Hospital staff did not interfere with the survey, so the objectivity and authenticity of the respondents' answers can be ensured. " And "Questionnaires with missing answers, or too short filling time, were deemed invalid. " (See Lines 230-233 for details). These are also to ensure the objectivity of the questionnaire survey.

  • For OD cost matrix analysis and Statistical analysis

After the revision, we added the detailed content of OD cost matrix model analysis and statistical analysis (including Chi square test, correspondence analysis and regression analysis) in Methodology section. The relevant words include the descriptions of the methods as well as all the procedures performed, to ensure the objectivity of the analysis. See Lines 252-287 and Lines 346-382 for details.

In addition, the results obtained by these methods have been scientifically and quantitatively calculated. So the results are also objective.

  1. Could you plase clarify why did you do a second phase of surveys by phone?

The responses:

Telephone follow-up interview was conducted to find out what accessibility factors influencing the elderly’s decision on whether or not to choose the nearest hospital.     

Therefore, we need to know the accessibility evaluations of the nearest hospital by the elderly. Then, taking the evaluations of accessibility factors as the independent variables, and ‘whether or not to choose the nearest hospital’ as the dependent variable, a statistical analysis can be made.

However, the information about whether or not the elderly choose the nearest hospital was not available in the first stage of questionnaire survey. For these elderly people, they lack quantitative knowledge to accurately identify whether the hospital that they went to was the nearest. Hence, the OD cost matrix analysis was needed. (Lines 248-251)

Then, after knowing the nearest hospital for each elderly person, the researchers interviewed him/her again by phone. Older adults who chose the nearest hospital were asked to evaluate the accessibility of the hospitals they went to. For those seniors who did not choose the nearest hospital, they were asked to evaluate the accessibility of the nearest hospital, which obtained by OD cost matrix analysis (see Lines 289-297 for details). In this way, the accessibility factors of the nearest hospitals that encourage or hinder their access would be explored.

  1. Exclusion and Inclusion criteria are not clearly defined. Also, could you please explain why did you exclude the patients from other regions?

The responses:

After revision, we have clearly defined the exclusion and inclusion criteria for samples. Such as how respondents were selected at each hospital, which questionnaires were deemed invalid, and which respondents were excluded from the study. Details for numbers of questionnaires distributed, excluded and included in each hospital have also been supplemented. See Lines 229-241 and Appendix Table A2 for details.

Reasons for exclusion can be seen in Lines 233-239: “According to the survey, it was found that some elderly people were from areas outside Hefei urban city and even Anhui Province. For these elderly people, the reasons for selection of hospitals were more complicated, while accessibility of the hospital was generally not one of them (Zhan & Fu, 2022; Pan, 2006). Since this study gives a discussion on the elderly's choice of hospitals from the perspective of accessibility, such data is not helpful for the survey and should be excluded.

  1. Figure 4 would benefit of a more objective description.

The responses:

We have revised the description of Figure 4. See Lines 416-450 for details.

Reference for the response:

  1. Yang, S.; Wang, D.; Li, C.; Wang, C.; Wang, M. Medical Treatment Behaviour of the Elderly Population in Shanghai: Group Features and Influencing Factor Analysis.  J. Environ. Res. Public Health2021, 18, 4108. https://doi.org/10.3390/ijerph18084108
  2. Jing, X.; Xu, L.; Qin, W.; Zhang, J.; Lu, L.; Wang, Y.; Xia, Y.; Jiao, A.; Li, Y. The Willingness for Downward Referral and Its Influencing Factors: A Cross-Sectional Study among Older Adults in Shandong, China.  J. Environ. Res. Public Health2020, 17, 369. https://doi.org/10.3390/ijerph17010369
  3. Chu, F.-Y.; Chang, H.-T.; Shih, C.-L.; Jeng, C.-J.; Chen, T.-J.; Lee, W.-C. Factors Associated with Access of Marital Migrants and Migrant Workers to Healthcare in Taiwan: A Questionnaire Survey with Quantitative Analysis.  J. Environ. Res. Public Health2019, 16, 2830. https://doi.org/10.3390/ijerph16162830
  4. Yan, Y.-H.; Kung, C.-M.; Yeh, H.-M. The Impacts of the Hierarchical Medical System on National Health Insurance on the Resident’s Health Seeking Behavior in Taiwan: A Case Study on the Policy to Reduce Hospital Visits. J. Environ. Res. Public Health 2019, 16, 3167. https://doi.org/10.3390/ijerph16173167
  5. Cheng, M.; Tao, L.; Lian, Y.; Huang, W. Measuring Spatial Accessibility of Urban Medical Facilities: A Case Study in Changning District of Shanghai in China. J. Environ. Res. Public Health 2021, 18, 9598. https://doi.org/10.3390/ijerph18189598
  6. Rao, Y.; Zhong, Y.; He, Q.; Dai, J. Assessing the Equity of Accessibility to Urban Green Space: A Study of 254 Cities in China. J. Environ. Res. Public Health 2022, 19, 4855. https://doi.org/10.3390/ijerph19084855
  7. Wan, J.; Zhao, Y.; Chen, Y.; Wang, Y.; Su, Y.; Song, X.; Zhang, S.; Zhang, C.; Zhu, W.; Yang, J. The Effects of Urban Neighborhood Environmental Evaluation and Health Service Facilities on Residents’ Self-Rated Physical and Mental Health: A Comparative and Empirical Survey. J. Environ. Res. Public Health 2022, 19, 4501. https://doi.org/10.3390/ijerph19084501

Reviewer 2 Report

General:

The authors could conceptualize the main variables as preference and actual behavior (or ‘revealed preference’, Samuelson), and the gap between them.

A missing analysis is a regression including as independent variables importance of distance, N of transfers, convenience of public transportation, congestion.

Under introduction.

The introduction and the literature review are repetitive and should be streamlined; it needs editing. For example, the research question is posed more than once, the considerations are mentioned several times in several locations, though never adequately discussed in terms of previous literature.

Mention the Anderson model early in the review of the literature and which variables from the model are relevant to the current analysis. Then, first mention the consideration found in the literature which are universal and only then describe the Chinese circumstances, and the specific location of the area, when different (they usually align with the global literature.

the behaviour of nearby medical treatment is..” there is no such behavior; maybe the authors meant “choosing nearby medical treatment is a behavior....”.

Do not use words of causation (“affect”, “influence”) as no experiment was conducted. This is a cross sectional study. There is no prediction, either. Remove these words throughout the manuscript and stay with associations.

“principles” would better be substituted by “considerations”.

Change “intends to address”  to “addressed”.

I cannot see how the research question “Is the proximity of the medical service facilities (such as general tertiary hospitals) essential to the elderly?“ can be answered. Maybe change to how high is it evaluated among the many other considerations.

The description of the sections (ll. 89-92) is redundant, as the sequence mostly follows standard scientific writing.

The location of describing Hefei is not optimal. Include it as a supplement or think of another option.

Under literature review.

Include it as part of the introduction.

“Academic circles” – change into “academic literature”.

Change “reflect” into “operationalize”.

Under Methodology.

It is a pity the authors did not measure the perceived standard/expertise of hospitals. Pls include it under limitations.

All abbreviations first need to be written in full (e.g., OD (origin destination), even GIS).

Do not use “etc.” when describing measures. Describe in full detail. If space is a problem, include a supplement.

In Table 1, drop “of samples” (in N), substitute “attribute with “values”. Pls specify the units of the income (I assume Chinese), and include a conversion value.

Under Results

Table 3 is too detailed. A mean and SD in the text would do.

There are two Table 4’s.

Figure 5 is difficult to interpret. If it’s just descriptive, it can be moved to an appendix.

L 512 – moderate the phrase “nothing to do” to “little to do”.

Under discussion.

The authors write that proximity is the “traditional notion”. This was not established in the introduction, as no other variables are fully reviewed. There was no comparison between associated variables and the choice of care. Also, I doubt any administrator “blindly increases the number of medical facilities”. Hospitals are very expensive and constructing a hospital is done after considerable planning.

L 464 – change the wording in the  “following research is carried out”.

The manuscript needs language editing.

Author Response

Comments of Reviewer 2 and responses from the authors

Dear professor,

Thank you very much for your help in processing the review of our manuscript entitled "Analysis of the Elderly’s Preferences for Choosing Medical Service Facilities from the Perspective of Accessibility: A Case Study of Tertiary General Hospitals in Hefei, China" (ijerph-1768250).

After the revision, all concerns/suggestions you raised have been addressed as follows:

General:

  1. The authors could conceptualize the main variables as preference and actual behavior (or 'revealed preference', Samuelson), and the gap between them.

The responses:

Thank you for your help. With your suggestion, relevant words have been conceptualized as preference, actual behavior and the gap.  

For example, “In order to analyze the preference for hospital distance among the elderly with different attributes…..”(Lines 350-351),  “…..regression analysis was performed again to finally identify the accessibility factors related to the elderly’s actual behaviors”(Lines 380-382), “They had to compromise according to the actual conditions, leading to a gap between their preference and actual behavior”(Lines 459-461),  and the title and variables of Table 5.

  1. A missing analysis is a regression including as independent variables importance of distance, N of transfers, convenience of public transportation, congestion.

The responses:

The regression analysis has been added, as shown in Lines 375-382, Lines 483-509 and Table 7.

Under introduction.

1.The introduction and the literature review are repetitive and should be streamlined; it needs editing. For example, the research question is posed more than once, the considerations are mentioned several times in several locations, though never adequately discussed in terms of previous literature.

The responses:

Thank you for your suggestion. We have simplified the Introduction and Literature review. And the discussions of Literature review have been moved into the Introduction, referring to your suggestion below.

The problem that “the research question is posed more than once, the considerations are mentioned several times in several locations”, has been revised. The discussion of research hypothesis, aims and research questions is included in Section 1.2.  Research problems that had appeared elsewhere were deleted.

  1. Mention the Andersen model early in the review of the literature and which variables from the model are relevant to the current analysis. Then, first mention the consideration found in the literature which are universal and only then describe the Chinese circumstances, and the specific location of the area, when different (they usually align with the global literature.

The responses:

Admittedly, Anderson model is a very common and effective model, which is often used as a theoretical framework to analyze the influencing factors of medical utilization behavior. Predisposing characteristics, Enabling resources and Need are the influencing factors of health service utilization; together they form the initial structure of Anderson's model. Many scholars use it for empirical research on medical treatment, with proper selection of measurement indicators.

However, the purpose of the sentences where Anderson model appeared was to indicate that " gender, age, and education level are important determinants of emergency medical use by the elderly after accounting for predisposing factors, demands, and other relevant factors" (Lines 60-63). And the whole paragraph was to explain that " the social factors of the elderly, including their residence, household registration type (rural or urban residents), family structure, gender, age, and education background, are associated with their behaviors in seeking medical treatment " (Lines 63-66) .  

In fact, Anderson model is not our study object, nor is it what we want to emphasize.

After your reminding, we found that the appearance of Anderson model here would confuse our research object. Therefore, we have deleted the words "Anderson model" and its relevant introduction.

  1. "the behavior of nearby medical treatment is... " there is no such behavior; maybe the authors meant "choosing nearby medical treatment is a behavior... "

The responses:

Thank you for your reminder. These errors have been revised.

  1. Do not use words of causation ("affect", "influence") as no experiment was conducted. This is a cross sectional study. There is no prediction, either. Remove these words throughout the manuscript and stay with associations.

The responses:

The words “affect", "influence" have been replaced. For instance, the words “Influencing factors of nearby medical treatment behaviour”(Line 511 of the unrevised manuscript) have been changed to “Association between hospital accessibility factors and elderly behavior”(Line 482 of the revised version).

  1. "principles" would better be substituted by "considerations".

The responses:

We have revised it as you suggested. See Line 123 for details.

  1. Change "intends to address" to "addressed".

The responses:

We have revised it (Line 129).

  1. I cannot see how the research question “Is the proximity of the medical service facilities (such as general tertiary hospitals) essential to the elderly?" can be answered. Maybe change to how high is it evaluated among the many other considerations.

The responses:

This sentence was revised to "How highly do the elderly evaluate the distance to tertiary general hospitals in relation to other considerations?" See Lines 129-130 for details.

  1. The description of the sections (II. 89-92) is redundant, as the sequence mostly follows standard scientific writing.

The responses:

The description has been deleted.

  1. The location of describing Hefei is not optimal. Include it as a supplement or think of another option.

The responses:

Thank you for your advice. Now we have put the description into the Methodology (which was also suggested by another reviewer). See Lines 137-160 for details.

Under literature review.

  1. Include it as part of the introduction.

The responses:

The Literature review has been incorporated into the Introduction. See Lines 43-85 for details.

  1. “Academic circles" - change into “academic literature" .

The responses:

The words have been revised (Line 82).

  1. Change “reflect" into “operationalize".

The responses:

The words have been revised. See Line 83.

Under Methodology.

  1. It is a pity the authors did not measure the perceived standard/expertise of hospitals. Pls include it under limitations.

The responses:

Thank you for your advice. We have added evaluation standard regarding facility allocation and staff expertise of tertiary hospitals set by Anhui Provincial Health Commission. See Lines 163-166 and Appendix Table A1 for details. Based on the standard, the tertiary hospitals were rated by Hefei Municipal Government.

  1. All abbreviations first need to be written in full (e.g., OD (origin destination), even GIS).

The responses:

We have made these revisions. See Line 17 (for OD), Line 210 (for GIS) and Lines 209-210 (for SPSS).

  1. Do not use “etc." when describing measures. Describe in full detail. If space is a problem, include a supplement.

The responses:

We have deleted “etc.” and described all the details. For example, “……the items in questionnaire also included each participant's residence location, and the subjective concern for hospital distance, etc” (Lines 270-271 of the unrevised version), has been modified into

 “The questionnaire contained information about the basic attributes of gender, age, household registration type, number of people living at home, number of generations living at home, education level, original occupation, number of children, household per capita annual income, type of medical insurance, travel activity status and number of private cars owned. In addition, the items also included each participant's residence location, and their preference for hospital distance.”  (Lines 241-247 of the revised version).

  1. In Table 1, drop “of samples" (in N), substitute “attribute with “values". Pls specify the units of the income (I assume Chinese), and include a conversion value.

The responses:

According to your suggestion, we have revised Table 2 (At the suggestion of another reviewer, the original Table 1 has been moved to the next section. This table is now Table 2).

Under Results

  1. Table 3 is too detailed. A mean and SD in the text would do.

The responses:

As your suggestion, we have modified these words; see Lines 385-392 and Table 3.

  1. There are two Table 4's.

The responses:

Before the revision, Table 4 was set to repeat header row. The table was split into two pages that looked like two tables (but are the same table). After revision, we have removed the "repeat header row" setting from all tables to avoid misunderstanding.

  1. Figure 5 is difficult to interpret. If it's just descriptive, it can be moved to an appendix.

The responses:

Figure 5 has been moved to the appendix as Figure A1.

  1. L 512 - moderate the phrase “nothing to do" to “little to do".

The responses:

These words have been revised. See Line 483.

Under discussion.

  1. The authors write that proximity is the “traditional notion". This was not established in the introduction, as no other variables are fully reviewed. There was no comparison between associated variables and the choice of care. Also, I doubt any administrator “blindly increases the number of medical facilities". Hospitals are very expensive and constructing a hospital is done after considerable planning.

The responses:

The words "traditional notion" and "blindly increases the number of medical facilities" were the inaccuracy in our language expression. These words have been removed. Now the sentences have been modified to

Overall, the findings of this study have clarified the rules of the elderly's medical behavior and the factors considered when choosing a hospital. And instead of shortening the distance between hospitals and residential areas to encourage the elderly to seek medical treatment nearby, it would be more effective to address issues such as traffic congestion, inconvenient platform settings, and the high number of transfers required for public transportation.” See Lines 586-591 for details.

  1. L 464一change the wording in the “following research is carried out".

The responses:

The words "following research is carried out" (L464 in the unrevised manuscript) are redundant and have been delated. Now the sentence has been modified to “To explore the characteristics of the elderly who chose the nearest medical treatment, a chi-squared analysis of their attributes and behavior was conducted.” See Lines 467-469 for details.

  1. The manuscript needs language editing.

The responses:

Language editing has been made. The proof is attached.

Reviewer 3 Report

I was invited to revise the paper entitled "Analysis of the Elderly’s Preferences for Choosing Medical Service Facilities from the Perspective of Accessibility: A Case Study of Tertiary General Hospitals in Hefei, China". It is a case study about the healthcare services organization of the Chinese Province of Anhui. Authors tried to evaluate if the neighbourhood can influence the access to hospital among elderly patients.

The topic is interesting and faced off an important topic for public health.

I have some observations:

- Authors should re-organize the paper structure: lines 89-124 should be shifted to methods section. In addition, the literature review is part of study backround so it should be presented in the introduction section. Statistical analysis should be shifted in methods section. Table 1 reports results so it cannot be reported in methods;

- Authors should better describe how elderly residents were enrolled and selected;

- Sample size estimation was totally lacking;

- Statistical analysis section should be improved. All test performed should be properly described. Actually, analysis cannot be evaluated;

- Among Discussions, strenght and limitation section should be added.

Author Response

Comments of Reviewer 3 and responses from the authors

Dear professor,

Thank you very much for your help in processing the review of our manuscript entitled "Analysis of the Elderly’s Preferences for Choosing Medical Service Facilities from the Perspective of Accessibility: A Case Study of Tertiary General Hospitals in Hefei, China" (ijerph-1768250).

After the revision, all concerns/suggestions you raised have been addressed as follows:

  1. Authors should re-organize the paper structure: lines 89-124 should be shifted to methods section. In addition, the literature review is part of study background so it should be presented in the introduction section. Statistical analysis should be shifted in methods section. Table 1 reports results so it cannot be reported in methods;

The responses:

Thank you for your suggestions.

The description in Lines 89-93 of the unrevised manuscript is redundant, as the sequence described mostly follows standard scientific writing. Therefore, it has been deleted after the revision (which was suggested by another reviewer).

According to your suggestion, Lines 94-124 of the unrevised version have been moved to the Methodology. See Lines 137-160 of the revised version for details.

The Literature review has been incorporated into the Introduction. See Lines 43-85 for details.

Statistical analysis has been added in the Methodology (Lines 346-382).

Table 1, as you suggested, has been put in Section 4 (Result and analysis). The table is now Table 2.

  1. Authors should better describe how elderly residents were enrolled and selected;

The responses:

After revision, we have added the details of the elderly selection and the questionnaire survey process. See Lines 229-241 for details.

  1. Sample size estimation was totally lacking;

The responses:

We have added the description of sample size estimation:

The sample size for the one-to-one interviews conducted in 2018 was determined by referring to the number of visits by the elderly to tertiary general hospitals in Hefei urban city in the previous year. According to 2017 statistics from the Hefei Municipal Statistics Bureau and Hefei Municipal Health Commission, there were 2.75 million annual visits by the elderly to these hospitals (Hefei Municipal Bureau of Statistics, 2018). The formula of sample size estimation is

                             (1)

In the formula, N is the total visit number of the elderly (2.75 million), e refers to the margin of error (5%), p is the response distribution (50%), confidence level is 95%, and z-score is 1.96 (Li, Liu &Liu, 2020; Wu, 2010). A minimum sample size of 384 patients was calculated.

See Lines 219-229 for details.

  1. Statistical analysis section should be improved. All test performed should be properly described. Actually, analysis cannot be evaluated;

The responses:

After revision, all statistical analyses performed have been described in detail. Such as:

  • For analyzing the preference for hospital distance among the elderly with different attributes, we have described in detail how to perform chi-square test first, and then select statistically significant variables for correspondence analysis. The setting of independent variables and dependent variables, and variable assignment, have been introduced.
  • In order to analyze the regularity of the elderly’s behaviors in seeking medical treatment, we performed chi-square test on the data. The process of the test has been introduced in detail.
  • In order to find out the factors associated with the elderly’s medical behavior, statistical analysis was conducted based on the data obtained from telephone interviews. Chi-square test has been performed first, followed by regression analysis. The process has been described in detail.

See Lines 346-382 for a detailed description of the above text.

All analysis results (including chi-square test, correspondence analysis and regression analysis) have been presented in Section 3.

  1. Among Discussions, strenght and limitation section should be added.

The responses:

The strength of the study can be seen in Lines 586-610 of the Discussion.

The limitation is in Lines 611-623.

Round 2

Reviewer 2 Report

The authors adequately addressed  my concerns

Reviewer 3 Report

Authors addressed all comments raised in the first round of review. 

In addition Authors should define which kind of regression was performed.